# Rheological Properties and Sensory Profile of Yoghurt Produced with Novel Combination of Probiotic Cultures

**DOI:** 10.3390/foods13193021

**Published:** 2024-09-24

**Authors:** Nebojša Ilić, Miona Belović, Nurgin Memiši, Mladenka Pestorić, Dubravka Škrobot, Lato Pezo, Rada Jevtić-Mučibabić, Yolanda Sanz, Jerome Brouzes

**Affiliations:** 1Institute of Food Technology in Novi Sad, University of Novi Sad, Bulevar cara Lazara 1, 21000 Novi Sad, Serbia; nebojsa.ilic@fins.uns.ac.rs (N.I.); mladenka.pestoric@fins.uns.ac.rs (M.P.); dubravka.skrobot@fins.uns.ac.rs (D.Š.); rada.jevtic@fins.uns.ac.rs (R.J.-M.); 2Imlek, Tolminska 10, 24000 Subotica, Serbia; nurgin.memisi@imlek.rs; 3Institute of General and Physical Chemistry, University of Belgrade, Studentski trg 12/V, 11158 Belgrade, Serbia; latopezo@yahoo.co.uk; 4Institute of Agrochemistry and Food Technology (IATA-CSIC), C/Catedrático Agustín Escardino Benlloch 7, 46980 Paterna, Valencia, Spain; yolsanz@iata.csic.es; 5Lallemand SAS, 19 Rue des Briquetiers, 31702 Blagnac, Cedex, France; jbrouzes@lallemand.com

**Keywords:** yoghurt, probiotic cultures, rheological properties, sensory analysis, fat content

## Abstract

Novel probiotic yoghurt was produced using the combination of bacterial cultures *Lactobacillus plantarum* HA119 and *Bifidobacterium animalis* subsp. *lactis* B94 and yoghurt bacteria *Lactobacillus delbrueckii* subsp. *bulgaricus* and *Streptococcus thermophilus*. Its basic nutritional composition, colour, texture, rheological properties, and sensory profile were compared with yoghurt produced using the same technological process and standard yoghurt cultures (control sample), as well as other commercially available yoghurts with different milk fat contents. Despite the fat content of the yoghurt made with the new probiotic cultures being 1.44%, its apparent viscosity was similar to that of high-fat yoghurt (2.99%). Other results from rheological measurements indicate that the new yoghurt had a stronger protein network, presumably due to the higher number of exopolysaccharides compared to both control and commercial yoghurts. Sensory analysis revealed that there were no statistically significant differences between the novel probiotic yoghurt and high-fat yoghurt as perceived by panellists. In conclusion, this combination of probiotic cultures can be used to produce yoghurt with rheological and sensory properties similar to high-fat yoghurts, without the need for hydrocolloids or changes in the production process.

## 1. Introduction

Yoghurt is an ancient drink that has gained a lot of popularity in recent years due to its positive effect on human health [1,2]. It is produced by inoculating pasteurized milk with lactic acid bacteria *Lactobacillus delbrueckii* subsp. *bulgaricus* and *Streptococcus thermophilus*. There are various formulations and technological processes for yoghurt production, with each one resulting in different properties of the final product, including appearance, texture, and flavour [3]. Major factors that influence the sensory properties of yoghurt are milk composition, starter culture, heat treatment, incubation temperature, fermentation pH and cooling [4].

Texture is an important attribute of yoghurt quality and is closely linked to its rheological properties. These properties are influenced by a three-dimensional network of casein and denatured whey protein, which forms via disulphide bonding during pH reduction [5,6]. The thickness and consistency of yoghurt are associated with the total solids content, protein type and concentration, and the pH level, as a lower pH increases the strength of the protein gel [5,6]. Milk fat significantly affects the yoghurt’s gel structure due to fat globules [3]. With growing concerns about the health impacts of high-fat dairy products, there is a rising demand for low-fat (0.5% to 2% milk fat) or non-fat (<0.5% milk fat) alternatives [1]. Additionally, the choice of lactic acid bacteria strain can further influence yoghurt’s textural, rheological, and sensory properties, particularly if the strain produces exopolysaccharides (EPSs) [2,3]. EPSs can be classified as homopolysaccharides or heteropolysaccharides, depending on the number of different types of monosaccharide units in the structure, which can be presented as either a long linear chain or contain side chains [7]. The properties of EPSs produced by different lactic acid bacteria vary widely; they differ in terms of monosaccharide composition, branching, charge density, substitutions, and whether they are entirely excreted or stay attached to the cell wall [8]. EPSs can have both positive (lower syneresis, increased viscosity, and creaminess) and negative (the presence of so-called “ropy strains”) influence on the sensory properties of yoghurt [7,9]. Different EPSs can interact with casein during milk fermentation, influencing the formation of the three-dimensional protein network [10].

*Lactobacillus plantarum* HA119 and *Bifidobacterium animalis* subsp. *lactis* B94 are probiotic microorganisms whose biological activity has been studied extensively. The effect of these two strains on plasma bile acid profiles in healthy adults with obesity was studied by Culpepper et al. [11]. Treatment with *B. lactis* B94 resulted in higher plasma deconjugated bile acids, while the treatment with *L. plantarum* HA119 showed a trend towards higher plasma HDL concentrations compared to the placebo, indicating the potential effect of these bacteria on serum lipids. In another study, *L. plantarum* HA119 was part of a multistrain probiotic used for the supplementation of a high-protein diet in older women [12].

A study performed on rats demonstrated that *B. lactis* B94 can improve liver function and reduce intestinal damage [13]. *B. lactis* B94 in combination with inulin was shown to reduce the duration of acute watery diarrhea in children, especially the one caused by *Rotavirus* [14]. However, no significant benefit was obtained for alleviating constipation in adults with Prader–Willi syndrome [15] or in addition to the standard triple therapy for *Helicobacter pylori* eradication in children [16].

Despite the probiotic potential of *L. plantarum* HA119 and *B. lactis* B94, few studies have investigated the application of *B. lactis* B94 in yoghurt production [17,18]. In these studies, satisfactory survival of this probiotic culture was found after 21 and 28 days of storage, respectively. To date, there are no known studies focusing on yoghurt production with the *L. plantarum* HA119 strain. Additionally, it is important to note that combinations of probiotic strains can have either synergistic or antagonistic effects on yoghurt’s nutritional and sensory properties [19]. Therefore, this study aimed to investigate the impact of adding a combination of probiotic cultures *Lactobacillus plantarum* HA119 and *Bifidobacterium animalis* subsp. *lactis* B94 on the rheological and sensory properties of drinkable yoghurt prepared with standard yoghurt cultures in comparison with commercially available yoghurts. The study also analyzed and discussed the relationship between the physicochemical, rheological, and sensory properties of yoghurt.

## 2. Materials and Methods

### 2.1. Material

Preparation of the novel probiotic yoghurt (labelled as P3) began with milk standardized to 1.0% milk fat. The standardized milk was then pasteurized and homogenized at 95 °C and a pressure of 150 bar. After this, the milk was inoculated with *Lactobacillus plantarum* HA119 and *Bifidobacterium animalis* subsp. *lactis* B94, along with standard yoghurt cultures *Lactobacillus delbrueckii* subsp. *bulgaricus* and *Streptococcus thermophilus*. Following inoculation, fermentation was carried out at 42.5 °C for 3 h and 50 min. The pH value of the final fermented product was 4.75 at the beginning of cooling and 4.50 at the end of cooling. The yoghurt prepared using the same technological process but with only the standard yoghurt cultures (control sample) were labelled as P1. Other commercial yoghurt samples (P2, P4, and P5) were selected based on their milk fat content, ranging from 1% to 3%, because milk fat content is a major factor in consumers’ choice of yoghurt due to health concerns and/or sensory properties. Since two days are the average time that pass from the production of yoghurt to their placement on market shelves, yoghurts P1 and P3 were left in the refrigerator for two days after the production. Commercial yoghurts (P2, P4, and P5) were bought on the same day when analyses were performed.

### 2.2. Chemical Composition

The chemical composition of the yoghurts was determined according to the methods of the Association of Official Analytical Chemists [20] for dry matter, ash content, and protein (925.23, 945.46, and 991.22). Fat content was determined according to ISO 19662:2018 [21].

### 2.3. Colour Measurement

The colour of the yoghurt was directly measured using a Konica Minolta Chroma Meter CR-400, with three repetitions for each sample. The sample was placed in a 10 mm glass cell (CM-A98), which was then fixed in the specimen holder (CM-96). The CIE L* (lightness), CIE a* (red-green), and CIE b* (yellow-blue) colour parameters were measured using a D65 light source and an observer angle of 2°. The total colour difference (ΔE) between the samples was calculated using the following equation:(1)$$\Delta E=\sqrt{{\left({L^{*}}_{2}-{L^{*}}_{1}\right)}^{2}+{\left({a^{*}}_{2}-{a^{*}}_{1}\right)}^{2}+{\left({b^{*}}_{2}-{b^{*}}_{1}\right)}^{2}}$$

### 2.4. Texture Analysis

Texture analysis of yoghurts was carried out using a TA.XT Plus Texture Analyser (Stable Micro Systems, Godalming, UK) in three replications. Firmness, consistency, cohesiveness, and the viscosity index were determined using a Back Extrusion Rig (A/BE), equipped with a 35 mm disc, an extension bar, and a 5 kg load cell. The test was carried out in a standard-size back extrusion container (50 mm diameter), approximately 75% full, after tempering for 30 min at room temperature (23 ± 2 °C). Compression mode was applied with the following instrumental settings: pre-test speed 1.0 mm/s; test speed 1.0 mm/s; post-test speed 1.0 mm/s; distance 30 mm; trigger force 5 g. Instrumental settings (project A/BE) were taken from the sample projects from the Texture Exponent Software (TEE32, version 6.1.6.0, Stable Micro Systems, Godalming, UK).

### 2.5. Rheological Analysis

The rheological properties of the yoghurts were analyzed with a Haake MARS rheometer (Thermo Scientific, Karlsruhe, Germany), with all measurements carried out in triplicates at 25 °C. The temperature during the flow curve measurements was regulated using the Phoenix II temperature control system which utilizes water circulation. The temperature during the yield stress and dynamic oscillatory measurements was controlled by the Peltier plate system.

#### 2.5.1. Flow Curve Measurements

The flow curves were recorded in a shear rate range between 0 and 100 s^−1^. Each sample underwent three flow ramps (up, constant, and down), each lasting for 120 s, to evaluate the effect of thixotropy. Flow curve parameters were calculated by fitting the experimental data for the downward (descending) flow curve to the Ostwald–de Waele relationship using the software provided with the instrument (Equation (2)):(2)$$\tau =K{\dot{\gamma }}^{n}$$
where τ is the shear stress (Pa), $\dot{\gamma }$ is the shear rate (s^−1^), K is the consistency index (Pa s^n^), and n is the flow behaviour index. Apparent viscosity was obtained from the flow curve as a mean value at 100 s^−1^, during constant flow ramp. The thixotropic behaviour of the samples was evaluated by estimating the hysteresis loop area (*S*) between the upward (increasing shear rate) and downward (decreasing shear rate) flow curves.

#### 2.5.2. Yield Stress Determination

Stress-controlled (CS) measurements were carried out using a parallel plate sensor system (PP60) with a 60 mm diameter and a 1 mm gap to determine the yield stress of yoghurt samples. A logarithmic CS ramp was recorded over a shear stress of 0.1 to 200 Pa within 180 s. The intersection of the linear segments in a log–log plot of deformation versus stress values indicated the transition from elastic behaviour to flow and was considered as the yield stress value.

#### 2.5.3. Dynamic Oscillatory Measurements

Dynamic oscillatory measurements were performed using a parallel plate sensor system (PP35) with a 1 mm gap. Mechanical spectra (frequency sweeps) were measured over a frequency range of 0.1–10 Hz at a stress of 0.1 Pa, which was within the linear viscoelastic region as determined by a stress sweep test at 1 Hz.

### 2.6. Sensory Evaluation

A sensory descriptive analysis was conducted on yoghurt samples by a panel consisting of 12 trained sensory assessors, comprising 8 females and 4 males aged between 30 and 55 years. The assessors were trained according to ISO 8586:2023 [22], ensuring adherence to all safety protocols to safeguard the participants’ well-being. Detailed information about the study was provided to all participants, who then provided informed consent to partake in the analysis. The training sessions, consisting of three 1.5 h sessions, involved using various types of commercial yoghurt (*n* = 5) to aid in the selection and definition of attributes, familiarize assessors with the scale usage, and identify end anchors. The examination took place at the Accredited Sensory Laboratory of the Institute of Food Technology, University of Novi Sad, which was designed in accordance with international standards for test rooms [23] for sensory evaluation. The final list of descriptors was developed through a consensus approach under the guidance of an experienced panel leader, including definitions and assessment techniques (Table 1) to ensure an objective approach to sensory evaluation. The intensities of perceived yoghurt attributes were evaluated on a 100 mm linear scale anchored with descriptive words [24]. Yoghurt samples were removed from the refrigerator 30 min before testing to reach room temperature (23 ± 2 °C). The evaluation was conducted in individual sensory booths, where yoghurts were served in plastic cups coded with random 3-digit numbers. Water with low mineral content was used to clean the palate between the individual samples. The sensory evaluation followed a balanced factorial design, with the order of sample presentation determined by the experimental design for sensory analysis using XLSTAT-MX (XLSTAT 2018.7, Addinsoft, http://www.xlstat.com/, accessed on 15 July 2024).

### 2.7. Statistical Analysis

The data were statistically processed using the XLSTAT 2018.7 software package. All sensory data were presented as means ± standard deviation (SD). Variance equality was assessed using Levene’s test, and normal distribution was verified with the Shapiro–Wilk test. Analysis of variance (ANOVA) followed by Tukey’s honestly significant difference (HSD) post hoc test was employed to identify sensory attributes that significantly distinguished among samples and to analyze variations in the sensory profiles of the yoghurt samples. Descriptive sensory data were subjected to Principal Component Analysis (PCA), resulting in a sensory map used for positioning the assessor’s overall liking data [25].

Visual examination of the relationships among various colour, sensory, and texture parameters of yoghurt, including both sensory and instrumental measurements, was conducted through correlation analysis using R software version 4.0.3 (64-bit). PCA was employed to differentiate the yoghurt samples, using Statistica version 14.0.0.15 (Tibco Inc., Palo Alto, CA, USA, 2020; https://www.tibco.com/products/data-science, accessed on 15 July 2024).

## 3. Results and Discussion

### 3.1. Physicochemical Properties

The results of basic nutritional composition, as well as colour and texture measurements of analyzed yoghurts are presented in Table 2. Significant differences were observed in dry matter and ash content between the samples, with sample P5 having the highest levels of dry matter, ash, and fat. However, the differences in protein content between the samples were not significantly different. The pH values ranged from 4.17 to 4.39, which is common for this type of fermented milk product and is generally considered an indicator of good product quality [4].

The yoghurt samples differed significantly in measured colour, with the total colour difference ranging from 0.14 to 0.92 (Appendix A). These results indicate that differences were either not observable by the human eye (total colour difference < 0.5) or only slightly observable (0.5 < total colour difference < 1.5) [26]. The variations in yoghurt colour are attributed to the different biochemical activities of yoghurt microorganisms, which transform the casein complex from a micellar to a dispersed state [27,28]. Novel probiotic yoghurt (P3) was characterized by significantly highest firmness, consistency, cohesiveness and viscosity index, while other yoghurts had similar textural properties. To improve the textural properties of yoghurt, an increase in total solid, protein or fat content is typically applied [29]. In this case, it can be supposed that this network was reinforced by the presence of exopolysaccharides, as previous studies suggested that *Bifidobacterium* spp. can produce significant amounts of EPSs, which enhanced the overall firmness of the yoghurt [7,30].

### 3.2. Rheological Properties

From the point of view of colloid chemistry, yoghurts are concentrated dispersions of whey and casein protein particle aggregates that form a network trapping the serum phase and containing dispersed milk fat globules [3,31].

Several models have been applied to the results obtained from the flow curve recording (Figure 1). The Ostwald–de Waele equation (Equation (2)) showed the best correlation with experimental data and was therefore chosen to obtain the flow curve parameters, especially because yield stress was not detected in all yoghurt samples. All yoghurts exhibited typical non-Newtonian shear thinning behaviour (*n* < 1). Thixotropy was more pronounced in samples P3 and P5, indicating the presence of a fragile three-dimensional protein network that was completely destroyed during shearing [32]. The novel probiotic yoghurt (P3) had the highest yield stress values and the largest thixotropic loop area, suggesting a stronger protein network compared to other yoghurt samples. The specific strength of this protein network can be the result of electrostatic interaction between negatively charged EPSs and positively charged casein micelles since the pH values of yoghurts were below the isoelectric point of casein (Table 2), which is at pH 4.6 [33]. Although the apparent viscosity of P3 measured at 100 s^−1^ was similar to that of P5, sample P5 had a higher consistency index and more pronounced pseudoplasticity, as reflected in its significantly lower flow behaviour index. This can be explained by the higher fat content in sample P5 (2.99%). Namely, the greater presence of fat globules led to a smaller size of casein micelle aggregates and a higher total contact area of each aggregate, which resulted in a homogeneous network. During rheological measurements, the strength of this network is reflected in higher consistency indices of high-fat yoghurts compared to low-fat yoghurts [34].

Dynamic oscillatory measurements at low strain conditions (within the linear viscoelastic region) were performed to provide a better insight into the microstructure of yoghurts. The storage modulus (G′) is a measure of deformation energy stored in the sample during shearing and represents reversible (elastic) structural changes in the system. In contrast, the loss modulus (G″) is a measure of deformation energy dissipated during shearing and represents irreversible (viscous) structural changes in the system. Therefore, if G′ values are higher than G″ values at all frequencies, the material behaves as a viscoelastic solid. All examined yoghurt samples exhibited viscoelastic solid behaviour, with a slight increase in moduli with increasing frequency (Figure 2). This result is a characteristic of weak gels, and it is typical for yoghurts [35]. Control yoghurt (P1), novel yoghurt (P3), and high-fat yoghurt (P5) all showed more pronounced elastic and viscous components. G′ and G″ were previously shown to have a good correlation with the total solid content of yoghurts, which can explain the higher values obtained for samples P1 and P5 (dry matter 11.45% and 12.00%, respectively) [34]. However, high values of both moduli obtained for sample P3 can be explained only by the interaction of EPSs with the protein network by forming bridging links [33] since this sample had significantly lower dry matter content (9.72%).

### 3.3. Sensory Properties

Eleven sensory attributes, encompassing appearance, odour, flavour/taste, and texture/mouthfeel, were evaluated, and the ANOVA results for these attributes, including means and standard deviations, are presented in Table 3. Consistency was the sensory attribute that exhibited the highest variance among the samples, which corresponds to the results of textural and rheological measurements. It is interesting to notice that yoghurts with more prominent consistency (P3 and P5) had less prominent overall odour intensity. Other sensory attributes did not differ significantly between the yoghurt samples. Consistency was the sensory property that exhibited the highest variance among the samples, which corresponds to the results of textural and rheological measurements. It is interesting to note that yoghurts with more pronounced consistency (P3 and P5) had lower overall odour intensity. No significant differences were observed in the other sensory attributes among the yoghurt samples.

Additionally, a panel analysis was performed using a Mixed Models—Type III Sum of Squares analysis to assess the significance of different sources of variation (products, assessors, and the interaction between products and assessors). ANOVA of all dependent variables was conducted on all dependent variables according to the model:(3)Y= μ+P+A+P×A
where μ represents intercept, P represents Product (a fixed factor), while A (Assessor) and P × A are random factors. The results of the panel analysis are summarized in Table 4.

For each descriptor, the Type III sum of squares from the ANOVA for the selected model was calculated. The panel analysis in Table 4 revealed significant differences among the products for the following descriptors: white colour (F = 6.825, *p* = 0.000), creamy odour (F = 7.061, *p* = 0.000), consistency (F = 28.506, *p* < 0.0001), fat feel (F = 22.538, *p* < 0.0001), mouthcoating (F = 21.505, *p* < 0.0001), and overall flavour (F = 5.646, *p* = 0.001). Assessors showed significant variability for nearly all descriptors, particularly white colour (F = 18.988, *p* < 0.0001), aftertaste (F = 15.684, *p* < 0.0001), smoothness (F = 7.510, *p* < 0.0001), and homogeneity (F = 6.006, *p* < 0.0001). However, smoothness (F = 0.227, *p* = 0.922) and homogeneity (F = 0.566, *p* = 0.688) showed no significant product differences, while mouthcoating (F = 1.510, *p* = 0.162) and aftertaste (F = 2.227, *p* = 0.081) exhibited no significant assessor variability. The analysis revealed that the *p*-values for the descriptors’ smoothness and homogeneity were greater than 0.1, leading to their exclusion from further analysis. The panel analysis results indicated significant findings regarding various sensory aspects. Specifically, the fixed effect of Products, particularly white colour, was highly significant, demonstrating notable differences among the evaluated products. Similarly, the random effect of Assessors was also significant, highlighting variations in how different assessors evaluate products. However, the interaction term Products × Assessors was not statistically significant. On the other hand, both Products and Assessors significantly influenced overall odour, with noticeable differences between Products and varying assessments by different Assessors. This underscored the importance of considering both Product and Assessor effects when interpreting sensory evaluation results.

Significant effects were observed for creamy odour, where while product differences were not significant, evaluations by Assessors varied significantly. Similar findings were noted for consistency and fat feel, with substantial differences among products and variability in assessor evaluations. In contrast, aftertaste ratings were significantly influenced by the specific Assessor but not by product type. Overall flavour ratings were significantly impacted by both product and assessor effects. The F-value for Products indicates a significant influence on mouthcoating, whereas Assessors and the Products × Assessors interaction did not notably affect mouthcoating scores. For sour taste, the Product alone did not strongly influence perception, but the Assessor’s role was significant. Both Product and Assessor significantly contribute to overall flavour ratings, but their interaction did not notably affect the overall flavour perception.

#### Generalized Procrustes Analysis (GPA) and Principal Component Analysis (PCA)

Generalized Procrustes Analysis (GPA) is commonly employed in sensory analysis prior to minimizing scale effects and achieving a consensus configuration. It also facilitates the comparison of the terms used by different assessors to describe products.

GPA yields a consensus matrix reflecting the average product values across all assessors, offering a unified view of the product evaluations. This analysis highlights how individual assessor evaluations align with the reference product scores, revealing both consistencies and discrepancies in product assessments. Residuals calculated for each assessor measure deviations from the reference values, with high residuals indicating significant deviations and potential subjective bias, while low residuals suggest strong alignment with the consensus. The eigenvalues from the PCA provide insight into the variance explained by the principal components, with Factors F1 through F4 capturing the most significant variance in the product evaluations. Higher eigenvalues denote more important factors in understanding the variability of the data. By focusing on the top four factors, the analysis simplifies the data complexity while retaining critical information. The combination of product values, assessor residuals, and eigenvalues offers a comprehensive view of the product evaluation process, highlighting areas of agreement and disagreement among assessors.

Firstly, the residuals by-product after the scaling transformations revealed that product P2 had the smallest residual (2752.9), which suggests a likely consensus among the Assessors (Figure 3a). In contrast, the residual for product P4 (4035.8) was the highest, indicating that the Assessors probably did not reach a consensus regarding this product. These values reflected how well each object aligns with the common configuration derived from the analysis. P1 and P2 had the lowest residuals, suggesting they fit the model relatively well. In contrast, P4 and P5 exhibited higher residuals, with P4 showing the greatest deviation from the common configuration. This indicated that P4 and P5 deviate more significantly from the expected pattern compared to P1, P2, and P3, with P3 exhibiting a moderate fit.

Furthermore, the results of the GPA by assessors highlighted the degree to which each configuration aligns with the common pattern established by the analysis. Residuals for assessors are presented in Figure 3b. Assessors A1 and A2 have the lowest residuals (1011.04 and 659.26, respectively) indicating they fit the common configuration most accurately. In contrast, A5 has the highest residual (3600.81), suggesting it deviates significantly from the expected pattern (they gave scores that do not match the consensus). Other configurations, such as A3, A6, A7, A8, A9, A11, and A12, show moderate residuals, reflecting varying levels of alignment with the common configuration. Assessor A10, with the second-lowest residual (629.46), also fitted well but slightly less so than A1 and A2.

Figure 4 presents the scaling factors for the GPA transformations. A factor below 1 indicates that the corresponding assessor (Assessor) used a broader scale compared to others, while a factor above 1 suggests that the assessor used a narrower scale. It was noted that Assessors 5 and 11 used a narrower scale than the other assessors. A consensus test was conducted to verify the authenticity of the consensus configuration. This permutation test involved calculating the proportion of the original variance (Rc value = 0.53) explained by the consensus configuration, which was significantly higher than 95% of the values obtained through data permutation, for the four factor dimensions (F1–F4).

The GPA dimensions test results highlighted the strong influence of the first four factors on the data structure. F1, with the highest F-value of 198.35, is the most significant, indicating that it explained the largest portion of the variance. F2, F3, and F4 also contributed meaningfully, with F-values of 41.20, 34.56, and 66.69, respectively. The extremely low *p*-values (<0.0001) for all factors emphasized that these findings are not due to random chance. The quantiles, particularly the 100% for F1 and F2, underscored their critical role, while the slightly lower but still significant quantile for F3 (87.67%) suggested that it had a substantial, though slightly lesser, impact. These factors together provided a comprehensive explanation of the data variability.

Although the GPA includes a rotation step to align each configuration with the consensus configuration, the PCA plot represents the optimal transformation of the consensus configuration under standard PCA constraints. This PCA transformation is then applied to each Assessor’s configuration. The obtained eigenvalues for factors F1–F4 were 1752.6, 217.1, 167.1, and 111.4, respectively. The proportion of variability attributed to each axis was 78.0%; 9.7%; 7.4% and 4.6%, indicating that the variability was predominantly captured by the first two principal factors. When examining the variability distribution among the Assessors, the results were nearly identical for all of them.

According to GPA, a PCA biplot was created, for the sensory analysis of yoghurt samples (Figure 5). The first two principal components (PC1 and PC2) explained a high percentage of the total variability (87.62%). Based on the correlation analysis of descriptors obtained by Assessors during the sensory analysis of yoghurt, overall odour was predominantly placed on the left side of the graph, while other descriptors such as aftertaste, creamy odour, sour taste, white colour, overall flavour, consistency, fat feel and mouthcoating were more closely related to the positive direction of F1 coordinate.

Figure 5 shows that products P3 and P5 received the highest scores from assessors for descriptors such as aftertaste, creamy odour, sour taste, white colour, overall flavour, consistency, fat feel, and mouthcoating. The novel yoghurt sample (P3) was distinguished from others by its pronounced consistency, fat feel and mouthcoating, despite its lower fat content. Conversely, sample P5, with the highest fat content, was noted for its aftertaste, creamy odour intensity, sour taste and white colour. Additionally, product P4 received the highest score for overall odour.

According to the correlation analysis of products obtained by Assessors during the sensory analysis of yoghurt (Figure 6), a clear separation was observed for product P4. In contrast, the borders between products P1 and P2, as well as P3 and P5 were not evident. This indicated that the assessors consistently differentiated between P1 and P4, as well as P2 and P4, showing a consensus on these products. However, they did not clearly distinguish between P1 and P2, or between P3 and P5.

### 3.4. Correlation Analysis and Principal Component Analysis

Colour Correlation Analysis (CCA) was conducted using R software (version 4.0.3) to explore the similarities in the chemical, textural and rheological properties, as well as the sensory profile of probiotic yoghurt produced with a novel combination of probiotic cultures. The findings are presented graphically in Figure 7.

Several significant correlations were identified with a *p*-value less than 0.01 using Statistica software (version 14.0.0.15). Consistency, as determined by the sensory panel, was positively correlated with firmness (r = 0.990), consistency (r = 0.992), and apparent viscosity (r = 0.994), and negatively correlated with cohesiveness (r = −0.998) and the index of viscosity (r = −0.999). Since lower values of cohesiveness and the index of viscosity indicated higher cohesiveness and index of viscosity of the sample, these correlations indicated that the sensory panel’s assessment of yoghurt consistency aligned well with measurements obtained using a texture analyzer or rheometer. Good correlations between sensory-determined yoghurt consistency and measurements obtained by a texture analyzer were previously reported by Aktar [4]. Other textural properties assessed by the sensory panel that were correlated with instrumentally determined yoghurt properties (*p*-value < 0.05) included mouthcoating, which exhibited negative correlations with cohesiveness (r = −0.955) and the index of viscosity (r = −0.969) while showing a positively correlated with apparent viscosity (r = 0.966). The fat feel was also negatively correlated with cohesiveness (r = −0.965) and the index of viscosity (r = −0.979). The creamy mouthfeel is well-known to be related to the fat content of yoghurt, but it can be enhanced in low-fat yoghurts by the addition of agents that increase yoghurt viscosity [36], highlighting the connection between these quality parameters. The creamy odour was negatively correlated with cohesiveness (r = −0.979) and the index of viscosity (r = −0.982). The creamy odour was negatively correlated with cohesiveness (r = −0.979) and the index of viscosity (r = −0.982) but positively correlated with apparent viscosity (r = 0.979). Additionally, overall flavour was positively correlated with apparent viscosity (r = 0.983). These correlations indicated that more viscous yoghurt tended to have a more pronounced creamy odour and overall flavour, consistent with previous research [37]. Both white colour and homogeneity, as assessed by the sensory panel, were highly correlated (*p*-value < 0.05) with instrumentally determined colour parameters L* (lightness) and a* (negative values indicating a green tone), suggesting that brighter and less yellow yoghurts were perceived by panellists as whiter and more homogenous.

Textural properties of yoghurt determined by back extrusion (firmness, consistency, cohesiveness and index of viscosity) were highly correlated (*p* < 0.01) with apparent viscosity determined at a shear rate of 100 s^−1^, which was expected since both measurements were performed at high shear rates. Firmness, consistency and cohesiveness were also correlated (*p* < 0.05) with both storage (G′) and loss (G″) moduli, demonstrating similarities between these textural properties and elastic and viscous properties of yoghurt. An interesting very high correlation (r = 0.990) was observed between the consistency index determined by the rheometer and the colour parameters L* and a*, reinforcing the connection between yoghurt appearance and its rheological properties which were previously explained by solvation and aggregation of casein micelles presented in yoghurt [27].

Regarding the chemical properties, only dry matter and fat content of yoghurt showed a statistically significant correlation (*p* < 0.05) with sensory and physical properties, specifically smoothness and the b* (positive values indicating a yellow tone). Higher dry matter and fat content of yoghurt were associated with a decreased perception of smoothness by panellists and an increased yellow tone. A previous study showed a negative correlation between yellowness and the lack of lumps in yoghurt [27].

PCA was utilized to further explore the relationships between various samples. The proximity of points on the PCA plot signifies similarities in their patterns [38]. In the factor space, the direction of the vectors indicates the trends in the variables, while their lengths represent the strength of the correlations [39]. The first two principal components (PCs) accounted for a significant portion of the total variance in the dataset, explaining 86.57% in total. Specifically, the first principal component (PC1) contributed 64.89%, while the second principal component (PC2) accounted for 21.67% of the variance, as shown in Figure 8.

When analyzing the projection of variables onto the factor plane, significant negative contributions to the PC1 coordinate were observed for the sensory analysis parameters: white colour (5.04% of the total variance), creamy odour (4.83%), consistency (4.77%), fat feel (4.45%), mouthcoating (4.45%), and overall flavour (5.06%). The most positive influence among sensory analysis parameters on the PC1 coordinate was from homogeneity (4.57%). Negative influences of texture parameters on PC1 included firmness (4.90%), consistency (4.88%), apparent viscosity (4.99%), G′ (4.60%), and G″ (4.68%). Positive influences of texture parameters on the PC1 were noted for cohesiveness (4.82%) and the index of viscosity (4.75%). The PC2 coordinate was positively influenced by sensory analysis parameters such as smoothness (12.77%) and sour taste (8.0%), and by texture parameters including yield stress (9.08%), thixotropic loop area (6.63%), and flow behaviour index (9.51%). Negative influences on the PC2 coordinate included the colour coordinate b* (6.93%) and chemical parameters such as dry matter (11.26%), ash (5.03%), and pH value (4.08%).

Samples P1, P2, and P4 were characterized by high cohesiveness, an increased index of viscosity, and enhanced homogeneity and overall odour sensory profile parameters. Yoghurt P3 was characterized by high yield stress, apparent viscosity, consistency, and firmness. This sample was evaluated as having a sour taste, increased consistency, and a fatty mouthfeel. Yoghurt P5 was characterized by high apparent viscosity, increased consistency, firmness, and elevated values of moduli G′ and G″. It also had a high fat content, an augmented consistency index, and high values of colour coordinates L*, a*, and b*. Sample P5 was also noted for elevated scores in sensory analysis parameters, including consistency, creamy odour, mouthcoating, pronounced white colour, elevated overall flavour, and explicit aftertaste.

## 4. Conclusions

The study showed that adding a mixture of probiotic cultures, *Lactobacillus plantarum* HA119 and *Bifidobacterium animalis* subsp. *lactis* B94, to the standard yoghurt production process resulted in yoghurt with texture and mouthfeel similar to yoghurts with higher milk fat content. This result can be explained by the rheological measurements, which indicated that the probiotic mixture created a stronger protein network compared to the standard yoghurt with similar dry matter and fat content. The applied probiotic cultures may produce exopolysaccharides, which are negatively charged and can interact with positively charged casein at pH levels typical for commercially produced yoghurts, thus forming bridges which strengthen the network present in the yoghurt matrix. Sensory analysis revealed that there were no statistically significant differences between the novel probiotic yoghurt (1.44% fat) and the high-fat yoghurt (2.99% fat), suggesting that the new probiotic yoghurt can be readily accepted by consumers. Significant correlations between sensory evaluation and texture/rheological measurements were found, showing that changes in the texture of yoghurt can influence not only its mouthfeel but also its odour and colour. Therefore, the novel probiotic yoghurt can offer consumers both health benefits and improved sensory properties compared to standard low-fat yoghurt, but without the need for hydrocolloids or whey straining which are commonly used to improve the yoghurt texture.

## Figures and Tables

**Figure 1 foods-13-03021-f001:**
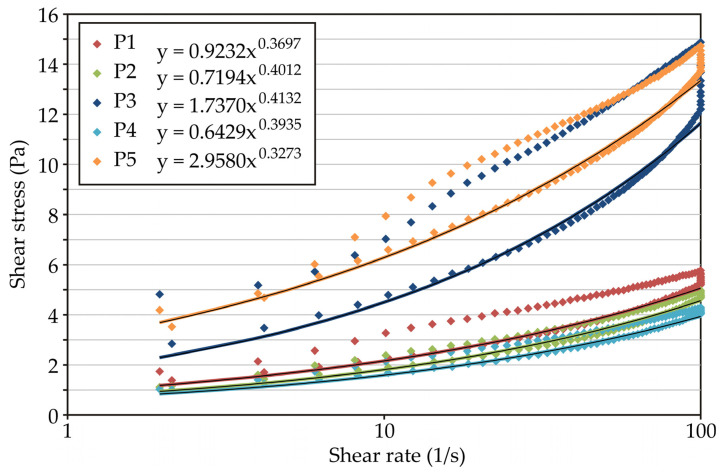
Flow curves of yoghurts; Ostwald–de Waele (power) model equations used for the fitting of the descending curves are presented in the legend.

**Figure 2 foods-13-03021-f002:**
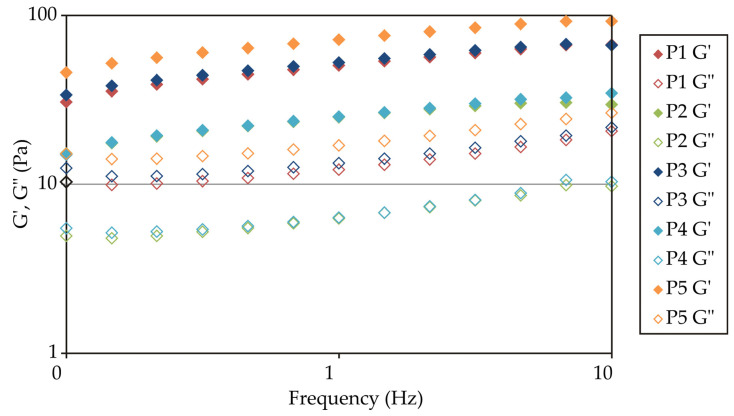
Mechanical spectra of yoghurts.

**Figure 3 foods-13-03021-f003:**
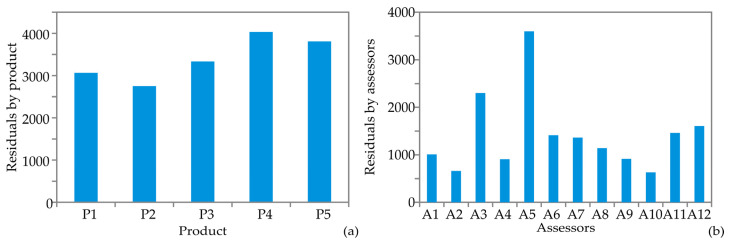
Residuals by (**a**) products and (**b**) assessors.

**Figure 4 foods-13-03021-f004:**
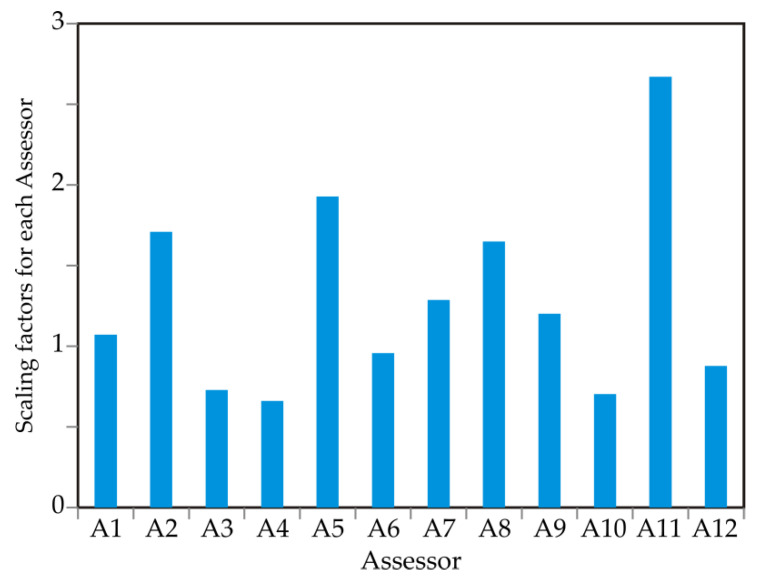
Scaling factors for Assessor’s scores.

**Figure 5 foods-13-03021-f005:**
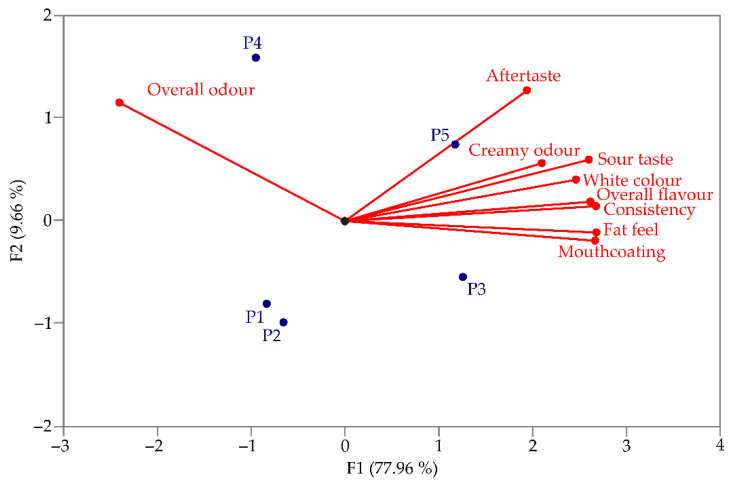
The PCA biplot diagram of the yoghurt sensory analysis.

**Figure 6 foods-13-03021-f006:**
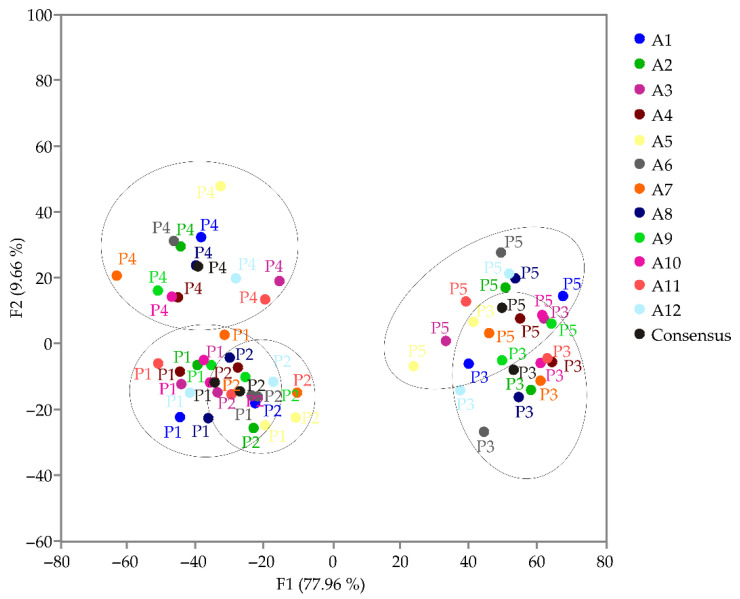
Correlation of products obtained by Assessors during sensory analysis of yoghurt.

**Figure 7 foods-13-03021-f007:**
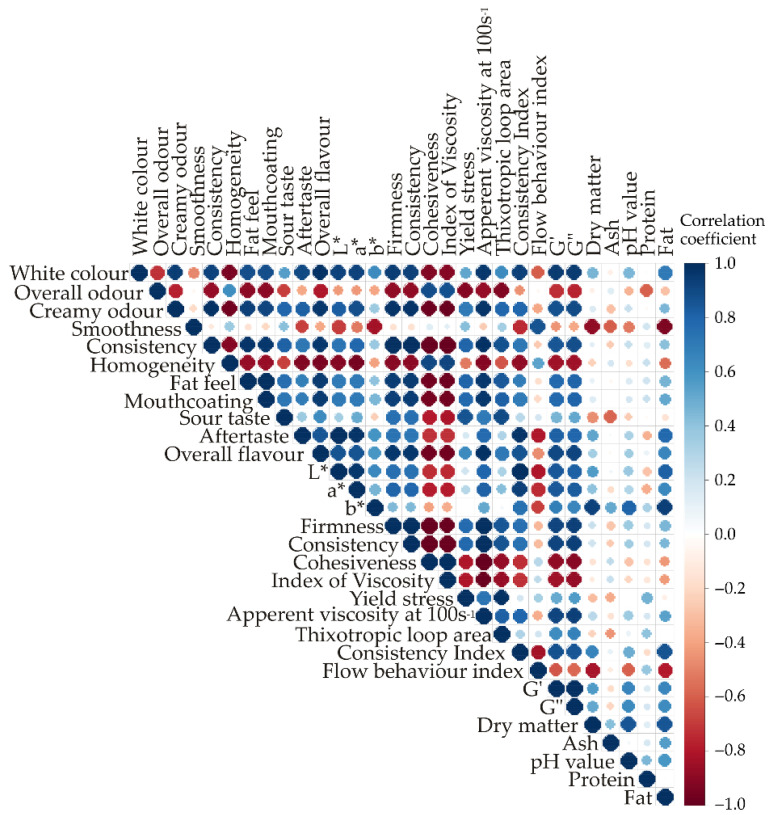
Colour correlation diagram between sensory analysis, colour, textural, rheological, and chemical parameters.

**Figure 8 foods-13-03021-f008:**
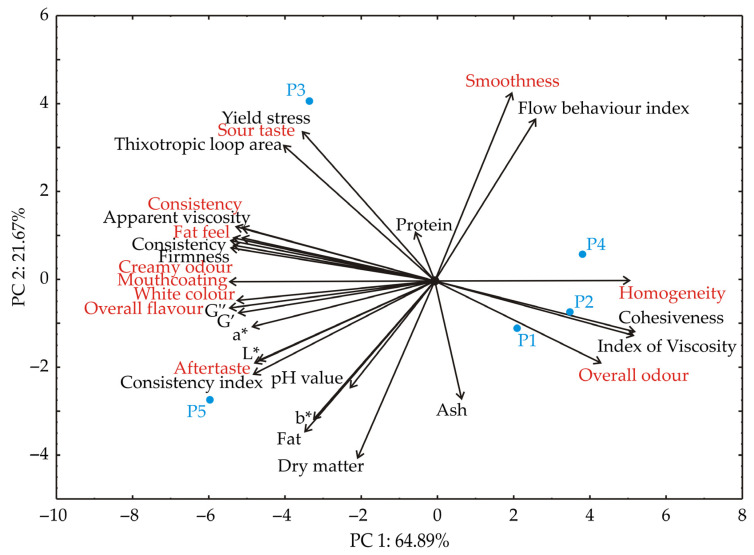
The PCA biplot diagram depicts the relationships among sensory analysis, colour, textural, rheological and chemical parameters.

**Table 1 foods-13-03021-t001:** The list of attributes used for sensory evaluation.

Attribute	Descriptor	Definition	Technique
Appearance	White colour intensity	Intensity of white colour	Visually
Odour	Overall odour intensity	Overall odour intensity of the sample	Olfactory
	Creamy odour intensity	Intensity of cream-like product odour	Olfactory
Texture	Smoothness	The degree to which the sample feels smooth and free of lumps	Orally
	Consistency	The degree of compactness of the sample when pressed between the tongue and palate	Orally
	Homogeneity	The degree of uniformity of the particles	Orally
	Fat feel	It refers to the intensity of the greasy feeling in the mouth during the manipulation of the sample between the tongue and the palate; the content of fatty components is observed	Orally
	Mouthcoating	Sensation of having a smooth/greasy coating on the tongue and other parts of the oral cavity	Orally
Taste	Sour	Basic taste produced by acid substances such as citric, acetic and lactic acid	Orally
	Aftertaste	Aftertaste intensity, 1 min after the sample is swallowed	Orally

**Table 2 foods-13-03021-t002:** Physicochemical properties of yoghurts *.

Parameter	P1	P2	P3	P4	P5
Dry matter (%)	11.45 ± 0.24 ^b^	10.68 ± 0.24 ^c^	9.72 ± 0.11 ^d^	9.73 ± 0.11 ^d^	12.00 ± 0.13 ^a^
Ash (%)	0.69 ± 0.02 ^bc^	0.78 ± 0.02 ^a^	0.67 ± 0.02 ^c^	0.68 ± 0.01 ^c^	0.73 ± 0.02 ^ab^
pH value	4.39 ± 0.12 ^a^	4.24 ± 0.06 ^ab^	4.25 ± 0.08 ^ab^	4.17 ± 0.06 ^b^	4.35 ± 0.07 ^ab^
Protein (%)	3.32 ± 0.14 ^a^	3.29 ± 0.20 ^a^	3.33 ± 0.18 ^a^	2.80 ± 0.22 ^a^	3.03 ± 0.15 ^a^
Fat (%)	1.73 ± 0.07 ^bc^	1.98 ± 0.14 ^b^	1.44 ± 0.09 ^cd^	1.22 ± 0.07 ^d^	2.99 ± 0.18 ^a^
L*	84.23 ± 0.09 ^c^	83.92 ± 0.01 ^d^	83.81 ± 0.09 ^e^	84.61 ± 0.01 ^a^	84.40 ± 0.01 ^b^
a*	−3.31 ± 0.02 ^c^	−3.31 ± 0.01 ^c^	−3.25 ± 0.04 ^b^	−3.07 ± 0.01 ^a^	−3.33 ± 0.01 ^c^
b*	7.33 ± 0.04 ^a^	6.77 ± 0.02 ^d^	6.83 ± 0.02 ^c^	6.53 ± 0.02 ^e^	6.98 ± 0.01 ^b^
Firmness (g)	26.18 ± 1.85 ^b^	24.78 ± 1.12 ^bc^	40.10 ± 1.92 ^a^	21.99 ± 0.68 ^c^	26.35 ± 2.03 ^b^
Consistency (gs)	684.0 ± 21.56 ^b^	669.4 ± 25.69 ^bc^	1117 ± 65.83 ^a^	594.5 ± 22.49 ^c^	691.4 ± 31.09 ^b^
Cohesiveness (g)	−18.53 ± 0.78 ^b^	−19.68 ± 0.80 ^b^	−66.14 ± 6.89 ^a^	−17.19 ± 0.54 ^b^	−19.48 ± 0.45 ^b^
Index of viscosity (gs)	−23.10 ± 4.66 ^b^	−25.10 ± 5.59 ^b^	−157.5 ± 14.25^a^	−9.70 ± 4.57 ^b^	−25.44 ± 4.15 ^b^

* Results are expressed as mean ± standard deviation of all replications. Values with the different superscripts within a row are statistically different (*p* < 0.05).

**Table 3 foods-13-03021-t003:** Sensory evaluation of yoghurt samples *.

Descriptor	P1	P2	P3	P4	P5
White colour intensity	55.5 ± 17.3 ^a^	53.2 ± 18.1 ^a^	61.4 ± 16.1 ^a^	53.2 ± 18.5 ^a^	67.2 ± 16.8 ^a^
Overall odour intensity	44.2 ± 23.9 ^ab^	44.8 ± 22.7 ^ab^	31.0 ± 16.1 ^b^	52.5 ± 23.7 ^a^	38.1 ± 17.7 ^ab^
Creamy odour intensity	30.4 ± 18.8 ^a^	26.9 ± 19.2 ^a^	49.2 ± 19.0 ^a^	31.8 ± 21.4 ^a^	51.8 ± 17.3 ^a^
Smoothness	77.2 ± 23.4 ^a^	76.8 ± 21.3 ^a^	79.3 ± 17.8 ^a^	78.2 ± 22.7 ^a^	74.3 ± 17.6 ^a^
Consistency	37.9 ± 10.1 ^b^	34.8 ± 16.2 ^b^	71.8 ± 16.8 ^a^	35.6 ± 15.1 ^b^	69.6 ± 14.4 ^a^
Homogeneity	80.6 ± 16.4 ^a^	80.8 ± 14.6 ^a^	76.4 ± 21.0 ^a^	79.1 ± 18.9 ^a^	74.0 ± 23.4 ^a^
Fat feel	31.0 ± 11.8 ^a^	38.6 ± 11.9 ^a^	64.9 ± 14.5 ^a^	30.6 ± 14.8 ^a^	62.7 ± 16.4 ^a^
Mouthcoating	33.2 ± 15.1 ^a^	42.7 ± 13.2 ^a^	68.6 ± 14.4 ^a^	31.2 ± 14.9 ^a^	68.1 ± 14.4 ^a^
Sour taste	34.7 ± 20.0 ^a^	34.0 ± 21.7 ^a^	56.3 ± 23.2 ^a^	42.5 ± 18.7 ^a^	44.2 ± 25.3 ^a^
Aftertaste	29.6 ± 28.5 ^a^	28.8 ± 23.5 ^a^	33.1 ± 27.6 ^a^	31.7 ± 29.6 ^a^	44.3 ± 34.1 ^a^
Overall flavour intensity	48.5 ± 17.2 ^a^	48.8 ± 22.5 ^a^	63.4 ± 15.8 ^a^	45.9 ± 18.8 ^a^	69.8 ± 10.6 ^a^

* Results are expressed as mean ± standard deviation of all replications. Values with the different superscripts within a row are statistically different (*p* < 0.05).

**Table 4 foods-13-03021-t004:** Panel analysis.

		Factors			
No	Descriptors	Products		Assessors	
		F	*p*	F	*p*
1	White colour	6.825	0.000	18.988	<0.0001
2	Overall odour	3.364	0.017	5.585	<0.0001
3	Creamy odour	7.061	0.000	4.047	0.000
4	Smoothness	0.227	0.922	7.510	<0.0001
5	Consistency	28.506	<0.0001	3.078	0.004
6	Homogeneity	0.566	0.688	6.006	<0.0001
7	Fat feel	22.538	<0.0001	2.371	0.021
8	Mouthcoating	21.505	<0.0001	1.510	0.162
9	Sour taste	2.533	0.054	2.206	0.032
10	Aftertaste	2.227	0.081	15.684	<0.0001
11	Overall flavour	5.646	0.001	2.311	0.024

F—F-values; *p*—*p*-values.

## Data Availability

The original contributions presented in the study are included in the article/supplementary material, further inquiries can be directed to the corresponding author.

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
