# Peer review of "Rheological Properties and Sensory Profile of Yoghurt Produced with Novel Combination of Probiotic Cultures"

_foods, 2024, doi:10.3390/foods13193021_

Round 1

Reviewer 1 Report

Comments and Suggestions for Authors

The paper submitted for review is a valuable work which concerns important issues associated with the demands of today’s consumers, who are increasingly interested in food with not only high nutritional value, but health-promoting value as well.

The description of the research material raises my doubts, there is no information on what basis the samples of commercial yoghurts were selected for testing, or only on the basis of fat content? Was their shelf life considered, how long after purchase the tests were carried out. This information is very important since during storage the acidity of yoghurts and its rheological properties may change and consequently sensory quality characteristics. Therefore, this is a very important factor. In turn, yoghurts P1 and P3 were most likely analyzed immediately after production, the work does not provide information on this subject, i.e. on what day after production they were analyzed. Were only 5 yoghurts analyzed? It would be good to know, how these parameters change during storage.

Due to the non-standardized research material and insufficient number of samples, it is not possible to draw clear conclusions.

Author Response

Response to Reviewer #1 Comments:

On behalf of all co-authors, I would like to express my thanks for all your valuable comments for the substantial improvement of the manuscript. Please find the detailed responses below and the corresponding revisions/corrections in track changes in the re-submitted files.

The paper submitted for review is a valuable work which concerns important issues associated with the demands of today’s consumers, who are increasingly interested in food with not only high nutritional value, but health-promoting value as well.

The description of the research material raises my doubts, there is no information on what basis the samples of commercial yoghurts were selected for testing, or only on the basis of fat content?

AUTHORS: The commercial yoghurts were chosen based on milk fat content and the popularity of drinkable yoghurt brands in Serbia. This was decided because milk fat content is a major factor in consumers’ choice of yoghurt due to health concerns and/or sensory properties. This explanation has been added to section 2.1. Material.

Was their shelf life considered, how long after purchase the tests were carried out. This information is very important since during storage the acidity of yoghurts and its rheological properties may change and consequently sensory quality characteristics. Therefore, this is a very important factor. In turn, yoghurts P1 and P3 were most likely analyzed immediately after production, the work does not provide information on this subject, i.e. on what day after production they were analyzed.

AUTHORS: Thank you for pointing this out, storage time is indeed an important factor for the rheological and sensory properties of yoghurt. Knowing this, in our study, yoghurts P1 and P3 were not analysed immediately after production. Instead, we allowed them to sit for 2 days in the refrigerator, as this is the average time from yoghurt production to being placed on market shelves. On the 3rd day, we purchased commercial yoghurts and performed analyses on the same day to minimize the impact on shelf life. We have, accordingly, added this information to section 2.1. Material.

Were only 5 yoghurts analyzed? It would be good to know, how these parameters change during storage.

AUTHORS: Only 5 yoghurts were analysed because they included, in addition to the control sample and the new probiotic yoghurt, the three most popular drinkable yoghurt brands in Serbia. As said before, this explanation has been added to section 2.1. Material. The change of yoghurt quality parameters during storage was out of the scope of this study, but we agree that it would be an interesting direction for further research.

Due to the non-standardized research material and insufficient number of samples, it is not possible to draw clear conclusions.

AUTHORS: This study deliberately used, besides a control sample, non-standardized samples taken from the market since the aim of the study was to compare the yoghurt produced with a novel combination of probiotic cultures with yoghurts already popular on the market. The results obtained in this way have practical significance and indicate to companies the market potential of the new product.

Reviewer 2 Report

Comments and Suggestions for Authors

This paper introduces the preparation of a new type of probiotic yoghurt, and compares it with the yoghurt prepared by standard culture and commercial yoghurt in terms of physical and chemical properties, texture, rheology, sensory properties. The experiment design is novel, but there are some problems at the same time, and it is suggested to major revision. Specific questions are as follows:

Major problems:

1.The major reason for the changes of physical and chemical properties of yogurt of the probiotic adding is the exopolysaccharide produced by probiotics. The author should provide some detail information about it.

2. The author wants to emphasize the physical and chemical properties, texture, rheology and sensory properties of the new probiotic yogurt. Therefore, in the introduction, the author should emphasize the exopolysaccharides produced by probiotics rather than the benefits of probiotics to human body.

3. In 3.2, the author mentioned that the reason for the different rheological properties of the new probiotic yogurt and the standard yogurt is the change in the structure of the protein network. The authors should give more references about above results.

4. In the panel analysis table in 3.3, the results obtained by the statistical analysis method chosen by the author cannot explain the conclusion well. For example, there are five products in the article, but only two assessors in the table; In 3.3.1, the authors do not show the values of the product, the assessor's residuals, and the factor F1-F4 eigenvalues in the graph or table, but it is mentioned in the discussion, the authors should provide more information about it.

5. In 3.3 and 3.4, the authors made PCA maps of sensory characteristics of yogurt, which were repetitive and the authors should explain why there are two PCA results and what’s the differences between them.

6. In 3.3 and 3.4, the author did not make a significance analysis. How did they get the P-value?

7. The author needs to add more information about the conclusions of protein network structure, and give some clear evidence about it is the protein instead of exopolysaccharide to contribute to the network structure in yogurt.

Comments on the Quality of English Language

Moderate editing of English language required.

Author Response

Response to Reviewer #2 Comments:

On behalf of all co-authors, I would like to express my thanks for all your valuable comments for the substantial improvement of the manuscript. Please find the detailed responses below and the corresponding revisions/corrections in track changes in the re-submitted files.

This paper introduces the preparation of a new type of probiotic yoghurt, and compares it with the yoghurt prepared by standard culture and commercial yoghurt in terms of physical and chemical properties, texture, rheology, sensory properties. The experiment design is novel, but there are some problems at the same time, and it is suggested to major revision. Specific questions are as follows:

Major problems:

1. The major reason for the changes of physical and chemical properties of yogurt of the probiotic adding is the exopolysaccharide produced by probiotics. The author should provide some detail information about it.

AUTHORS: We agree with this comment. The Abstract, Introduction, Results and Discussion and Conclusions sections were changed according to this remark, and new literature citations about the role of exopolysaccharides in yoghurt quality were added.

  1. The author wants to emphasize the physical and chemical properties, texture, rheology and sensory properties of the new probiotic yogurt. Therefore, in the introduction, the author should emphasize the exopolysaccharides produced by probiotics rather than the benefits of probiotics to human body.

AUTHORS: Thank you for this valuable suggestion. The part of the Introduction about the benefits of probiotics to the human body was shortened, and the part of the Introduction about exopolysaccharides produced by probiotics during yoghurt fermentation was expanded with new literature references.

  1. In 3.2, the author mentioned that the reason for the different rheological properties of the new probiotic yogurt and the standard yogurt is the change in the structure of the protein network. The authors should give more references about above results.

AUTHORS: The discussion in 3.2. Rheological properties was expanded to explain what caused the change in the structure of the protein network and new references were added. It was presumed that an interaction between negatively charged exopolysaccharides and positively charged milk proteins occurred, resulting in a stronger protein network, as observed in several previous studies.

  1. In the panel analysis table in 3.3, the results obtained by the statistical analysis method chosen by the author cannot explain the conclusion well. For example, there are five products in the article, but only two assessors in the table;

AUTHORS: Thank you for this observation. Table 4 has been reorganized in response to the Reviewer’s comments, now clearly displaying the factors investigated in the ANOVA analysis. The table includes both the Products (P1 – P5) and the Assessors (A1 – A12) as key variables. This rearrangement enhances the clarity of the data presentation, making it easier to interpret the influence of each factor on the sensory attributes being evaluated.

Besides Table 4, please check the text written in section 2.6. Sensory evaluation. As it was mentioned in the text, a sensory descriptive analysis was conducted on yoghurt samples by a panel consisting of 12 trained sensory assessors.

In 3.3.1, the authors do not show the values of the product, the assessor's residuals, and the factor F1-F4 eigenvalues in the graph or table, but it is mentioned in the discussion, the authors should provide more information about it.

AUTHORS: Thank you for this very important remark. Additional text was incorporated into the manuscript, primarily focusing on a general overview of GPA and a detailed discussion of the GPA results. This expansion enhances the manuscript by providing deeper insights into the methodology and the significance of the findings. Figure 3, which shows the residuals for products (a) and assessors (b), was also added to the manuscript.

  1. In 3.3 and 3.4, the authors made PCA maps of sensory characteristics of yogurt, which were repetitive and the authors should explain why there are two PCA results and what’s the differences between them.

AUTHORS: As was mentioned in the text, the first PCA biplot was created for the sensory analysis of yoghurt samples (Figure 5). The first two principal components (PC1 and PC2) explained a high percentage of the total variability (87.62%).

The second PCA analysis was carried out to introduce more insight to the relation between sensory analysis, colour, texture and chemical parameters of yoghurt samples (Figure 8). The first two principal components (PCs) accounted for a significant portion of the total variance in the dataset, explaining 86.56% in total.

In order to facilitate better differentiation between the two graphs, we revised the figure legends to provide more comprehensive information.

  1. In 3.3 and 3.4, the author did not make a significance analysis. How did they get the P-value?

AUTHORS: As was previously mentioned, the manuscript was expanded with additional text, mainly covering an overview of GPA and an in-depth analysis of the GPA results.

To perform a significance analysis in GPA using XLSTAT, the data was first prepared and standardized to remove scale effects. The GPA was then conducted by aligning the data from different sources to a common configuration through translation, rotation, and scaling. Residuals were computed for each configuration, and a permutation test with 300 permutations was applied to assess the statistical significance of the factors. XLSTAT generated F-values for each factor, such as 198.35 for F1 and 41.20 for F2, which were compared against a critical value of 2.54. The p-values, all less than 0.0001, indicated that these factors are highly significant, confirming that they contribute meaningfully to the common structure in the data.

The Colour Correlation Analysis (CCA) was conducted to explore the similarities in the chemical, textural and rheological properties, as well as the sensory profile of probiotic yoghurt produced with a novel combination of probiotic cultures. The findings are presented graphically in Figure 7. Several significant correlations were identified with a p-value less than 0.01, and they were listed in the text, in section 3.4. Correlation analysis and Principal Component Analysis.

  1. The author needs to add more information about the conclusions of protein network structure, and give some clear evidence about it is the protein instead of exopolysaccharide to contribute to the network structure in yogurt.

AUTHORS: Thank you for this very important suggestion. We have, accordingly, changed the Abstract, Results and Discussion section, and Conclusion section, to reflect the influence of both proteins and exopolysaccharides on the network structure in yogurt. The results of chemical, textural and rheological determinations indicate that the specific properties of new probiotic yoghurt originate from the electrostatic interaction between proteins and exopolysaccharides and do not represent the consequence of high fat or dry matter content.

Reviewer 3 Report

Comments and Suggestions for Authors

The article is not of sufficient quality to be published in this journal. Specific questions and points requiring attention are itemized below.

1. It is still difficult to determine the novelty of the work compared to what has already been published. What is the difference between what is published and what the authors want to publish? It is not clear.

2. Line 120. What temperatures were carried out in the rheological analysis?? Describe in detail.

3. Discussion section must be improved and compared with previous literature.

4. The quality of the Figures must be improved. Although the authors use excel as a plotter, these should be of better quality.

5. In Figure 1, why do samples have two curves with the same symbols? Explain in detail. 

6. In Figure 1, the axis must contain more points.

7. The authors must fit a mathematical model for Figure 1. 

8. In Figure 2, the authors must reach high frequencies (100 Hz). 

9. In Figure 1, the Shear rate (1/s) axis must be on the Log scale.

Author Response

Response to Reviewer #3 Comments:

On behalf of all co-authors, I would like to express my thanks for all your valuable comments for the substantial improvement of the manuscript. Please find the detailed responses below and the corresponding revisions/corrections in track changes in the re-submitted files.

The article is not of sufficient quality to be published in this journal. Specific questions and points requiring attention are itemized below.

  1. It is still difficult to determine the novelty of the work compared to what has already been published. What is the difference between what is published and what the authors want to publish? It is not clear.

AUTHORS: As Reviewer #3 remarked, there are numerous studies about the application of Lactobacillus plantarum and Bifidobacterium animalis subsp. lactis in yoghurt production, but those studies were performed with other strains of these bacteria. To the best of our knowledge, specific strains Lactobacillus plantarum HA119 and Bifidobacterium animalis subsp. lactis B94 have never been used in yoghurt production. These strains have been chosen for this study due to their potential health benefits which were described in the Introduction section, and the focus of the study was on the rheological and sensory properties of yoghurt since they are the most important for its potential industrial production.

  1. Line 120. What temperatures were carried out in the rheological analysis?? Describe in detail.

AUTHORS: Thank you for pointing this out. All rheological measurements were done at 25°C. The temperature during the flow curve measurements was regulated using the Phoenix II temperature control system which utilizes water circulation. The temperature during the yield stress and dynamic oscillatory measurements was controlled by the Peltier plate system. This explanation was added to the section 2.5. Rheological analysis.

  1. Discussion section must be improved and compared with previous literature.

AUTHORS: The Results and Discussion section was improved by a more detailed explanation of the obtained results and comparison with previous studies, as other Reviewers also suggested. All changes were marked by Track Changes in the Manuscript file.

  1. The quality of the Figures must be improved. Although the authors use excel as a plotter, these should be of better quality.

AUTHORS: The quality of the Figures was improved as much as possible when using Microsoft Excel to represent the results.

  1. In Figure 1, why do samples have two curves with the same symbols? Explain in detail.

AUTHORS: The samples have two curves with the same symbols because the first flow curve was recorded with an increasing shear rate (upward flow curve) and the second curve was recorded with a decreasing shear rate (downward flow curves) since both curves are needed to estimate the thixotropic behaviour of the yoghurt samples. This explanation was added to section 2.5.1. Flow curve measurements.

  1. In Figure 1, the axis must contain more points.

AUTHORS: More points were added to the y-axis in Figure 1.

  1. The authors must fit a mathematical model for Figure 1.

AUTHORS: The mathematical model (Ostwald-de Waele – power model) used for the fitting of the descending curve was added to Figure 1, and the explanation was added to Figure caption.

  1. In Figure 2, the authors must reach high frequencies (100 Hz).

AUTHORS: High frequencies (up to 100 Hz) were not used in this study, since trends of storage modulus (G’) and loss modulus (G”) change with frequency can be already seen at frequencies up to 10 Hz. For example, in one of the previously published papers on yoghurt in this journal, frequencies in the range from 0.1 Hz to 10 Hz were also used to study change of G’ and G” values:

https://doi.org/10.3390/foods11121764

  1. In Figure 1, the Shear rate (1/s) axis must be on the Log scale.

AUTHORS: Shear rate (1/s) axis was presented on the log scale in Figure 1.

Round 2

Reviewer 2 Report

Comments and Suggestions for Authors

This paper introduces the preparation of a new type of probiotic yoghurt, and compares it with the yoghurt prepared by standard culture and commercial yoghurt in terms of physical and chemical properties, texture, rheology, sensory properties, etc. The paper is full of novelty. The author has modified the paper, but there are still some problems, and it is suggested to make minor revisions. Specific questions are as follows:

Major problems:

1. In 3.3 and 3.4, the authors made PCA maps for the sensory characteristics of yogurt, and the contents were somewhat repetitive, suggesting that the authors could combine them.

2. In 3.4, the author did not make a significance analysis. How did he get the P-value? Authors need to be clear.

3. The author mentioned in the conclusion that "it interacts with positively charged milk protein", but the casein in yogurt has a negative charge, so the author needs to provide the pH condition of yogurt and the type of milk protein.

Minor problems:

1. The author mentions in 2.5.1 that the Ostwald-de Waele equation is formula 2, and in 3.2 that the Ostwald-de Waele equation is formula 1, which needs to be unified by the author

2. In Figure 7, products 1, 2 and 4 do not have P, so the author needs to modify them.

Comments on the Quality of English Language

Minor editing of English language required.

Author Response

Response to Reviewer #2 Comments:

This paper introduces the preparation of a new type of probiotic yoghurt, and compares it with the yoghurt prepared by standard culture and commercial yoghurt in terms of physical and chemical properties, texture, rheology, sensory properties, etc. The paper is full of novelty. The author has modified the paper, but there are still some problems, and it is suggested to make minor revisions. Specific questions are as follows:

AUTHORS: Thank you very much for your thoughtful comments and valuable feedback on our manuscript. We greatly appreciate your recognition of the novelty of our work. Your insights have been incredibly helpful, and we are confident that the revisions will enhance the clarity and impact of our findings. We have carefully considered the remaining issues you pointed out and will address them thoroughly to ensure the manuscript meets the highest standards. Please find the detailed responses below and the corresponding revisions/corrections in track changes in the re-submitted files.

Major problems:

  1. In 3.3 and 3.4, the authors made PCA maps for the sensory characteristics of yogurt, and the contents were somewhat repetitive, suggesting that the authors could combine them.

AUTHORS: Thank you for your insightful comment. As reviewer #2 observed, the results of sensory analysis were included in both PCA diagrams. However, they were drawn with different purpose and therefore cannot be combined. The initial PCA biplot was generated using Generalized Procrustes Analysis (GPA) with the XLSTAT 2018.7 software package to elucidate only the sensory characteristics of yoghurt samples (Figure 5). Given that sensory analysis provides a scientific evaluation of organoleptic properties, it is crucial to present PCA results for sensory attributes separately from those involving colour, textural, rheological and chemical parameters. Subsequently, a second PCA was performed to explore the relationships among sensory attributes as well as colour, textural, rheological and chemical parameters of the yoghurt samples (Figure 8). This PCA took into account all determined parameters and enabled grouping of yoghurt samples on the basis of both sensory and physico-chemical properties.

  1. In 3.4, the author did not make a significance analysis. How did he get the P-value? Authors need to be clear.

AUTHORS: The calculation of significant correlations was performed using Statistica 14.0.0.15, Tibco Inc., USA, 2020, and p-values below 0.01 were identified and discussed in section 3.4. Figure 7 was drawn using R software version 4.0.3 (64-bit) and it represents the Colour Correlation Analysis (CCA) results. The clear explanation was added to the text.

  1. The author mentioned in the conclusion that "it interacts with positively charged milk protein", but the casein in yogurt has a negative charge, so the author needs to provide the pH condition of yogurt and the type of milk protein.

AUTHORS: Thank you for this useful comment. Raw bovine milk contains casein with a negative charge, but during fermentation, the pH value of yoghurt drops below the isoelectric point of casein, which is at pH 4.6, causing the casein to become positively charged (Brüls et al., 2024; https://doi.org/10.1016/j.foodhyd.2023.109629). As can be seen in Table 2. Physicochemical properties of yoghurts, the pH value of all analyzed yoghurts was below the isoelectric point of casein, resulting in a positive charge of casein micelles. The text in the Results and Discussion section and Conclusion section was changed to describe this more clearly.

Minor problems:

  1. The author mentions in 2.5.1 that the Ostwald-de Waele equation is formula 2, and in 3.2 that the Ostwald-de Waele equation is formula 1, which needs to be unified by the author

AUTHORS: Thank you for this observation. Ostwald-de Waele equation is indeed the equation 2, and the text in the section 3.2. Rheological properties was corrected accordingly.

  1. In Figure 7, products 1, 2 and 4 do not have P, so the author needs to modify them.

AUTHORS: Thank you for the remark. We suppose that Reviewer #2 meant Figure 8, because we added one figure in the previous round of reviews. Figure 8 has been changed to include P labels added to 1, 2 and 4.

Reviewer 3 Report

Comments and Suggestions for Authors

The article can be accepted

Author Response

Response to Reviewer #3 Comments:

The article can be accepted.

AUTHORS: Thank you for the helpful suggestions that have significantly improved this scientific paper.